# ASSERT (Acute Sacral inSufficiEncy fractuRe augmenTation): randomised controlled, feasibility trial in older people

Terence Ong ,[1,2] Ana Suazo Di Paola,[3] Cassandra Brookes,[3] Avril Drummond ,[4] Paul Hendrick,[5] Paul Leighton,[6] Matthew Jones,[6] Khalid Salem,[7] Nasir Quraishi,[7] Opinder Sahota [1]

For numbered affiliations see end of article.

**Correspondence to**
Dr Terence Ong;
terenceong@doctors.org.uk

## ABSTRACT

**Objective** To determine the feasibility of designing and conducting a definitive trial to evaluate the effectiveness of sacral fracture fixation compared with non-surgical management among older people admitted with a lateral compression pelvic fragility fracture (PFF).

**Design** Single-site, parallel, two-arm randomised controlled feasibility trial.

**Setting** A UK tertiary centre hospital.

**Participants** Patients aged ≥70 years who were ambulating pre-injury requiring hospital admission (within 28 days of injury) with a type 1 lateral compression PFF.

**Interventions** The intervention group received sacral fracture fixation (cement augmentation±screw fixation) within 7 days of randomisation. Routine preoperative and postoperative care followed each surgical intervention. The control group received usual care consisting of analgesia, and regular input from the medical and therapy team.

**Primary and secondary outcome measures** The feasibility outcomes were the number of eligible patients, willingness to be randomised, adherence to allocated treatment, retention, data on the completeness and variability of the proposed definitive trial outcome measures, and reported adverse events.

**Results** 241 patients were screened. 13 (5.4%) were deemed eligible to participate. Among the eligible participants, nine (69.2%) were willing to participate. Five participants were randomised to the intervention group and four to the control group. The clinicians involved were willing to allow their patients to be randomised and adhere to the allocated treatment. One participant in the intervention group and two participants in the control group received their allocated treatment. All participants were followed up until 12 weeks post-randomisation, and had an additional safety follow-up assessment at 12 months. Overall, the proportion of completeness of outcome measures was at least 75%. No adverse events were directly related to the trial.

**Conclusions** There were significant challenges in recruiting sufficient participants which will need to be addressed prior to a definitive trial.

**Trial registration number** ISRCTN16719542.

### Strengths and limitations of this study

⇒ This feasibility study was designed to be pragmatic so that it could be delivered within current health-care setting.
⇒ The inclusion criteria mirrored the group of patients where there is uncertainty of the role for surgical intervention.
⇒ This feasibility study was unable to report on the effectiveness of surgical fixation for sacral fractures.

## INTRODUCTION

Pelvic fragility fracture (PFF) is common and its incidence rises exponentially with age peaking in those aged 85 years and over.[1–4] Among older adults, it is mostly caused by falls and bone fragility due to osteoporosis.[1 2] Recent years have also seen the annual incidence of PFF rising and the absolute number of patients with PFF hospitalised increased by 1.5–2 times.[2–4] The majority of these being older patients who require treatment in hospital to manage their pain and disability.[1 3]

The most common PFF identified involves the pubic rami of the anterior pelvic ring.[5 6] However, 55%–60% of these anterior pelvic ring fractures have concomitant involvement of the posterior ring, that is, a sacral fracture.[7 8] The sacrum is the triangular base of the spine below the lumbar vertebrae and forms the posterior part of the pelvic girdle.[9] Visualisation of sacral fractures on X-ray of the pelvis can be difficult.[10 11] Hence, many are diagnosed late when there is clinical suspicion of a more complex pelvic fracture.[9 11] Detection of posterior pelvic ring fractures is undertaken by either CT or MRI.[12 13] Such fractures that involve both the anterior and posterior part of the pelvic ring have worse outcomes. The average hospital length of stay for those with a combined anterior and posterior sacral

fracture was on average 2 weeks longer than those with just an isolated anterior ring pubic rami fracture[8]; 30% more patients lose their previous independence permanently and the rate of institutionalisation is also higher.[7]

The ultimate treatment goal for PFF is early restoration of mobility and function. This can only be achieved by effective and prompt pain relief. Fracture reduction and restoration of pelvic symmetry are less important. From a biomechanical point of view, an undisplaced anterior ring PFF is more stable than a posterior ring PFF. The pubic symphysis only contributes 15% towards pelvic stability compared with the posterior ring which provides the majority of the pelvis's structural support and stabilisation.[14] However, optimal pain control and early mobilisation remain challenging.[11 15] Around half of patients admitted with these fractures develop hospital and immobility complications.[4 6 8 16] One approach for treating such fractures is to stabilise the posterior ring fracture surgically and provide that potentially earlier pain relief, with a conservative, non-surgical approach for the more stable anterior pelvic ring fracture.

Surgical options for posterior ring fractures range from minimally invasive procedures, to open surgery with internal fixation.[17–19] Minimally invasive surgical techniques which involve percutaneous cement augmentation (injecting cement into the sacral ala at the side of the fracture) occasionally supplemented by a trans-sacral screw, also inserted using key-hole surgery, are increasingly being performed[20 21] and have been shown to reduce pain, reduce the amount of analgesia required postoperatively, increase patient mobility and are safe procedures in older people.[12 22–24] However, many of these studies were limited to observational and case–control studies which recruited a small number of participants and lacked a control arm.

A randomised controlled trial to evaluate the effectiveness of early surgical intervention for this type of pelvic fracture is required. Prior to conducting such a study, there remained uncertainty if such a trial could be delivered, the sample size required to determine its clinical effectiveness and the clinicians' adherence to allocated treatment groups. Hence, the aim of this present study was to determine the feasibility of a randomised controlled clinical trial of spinal sacral fixation (cement augmentation±screw fixation) compared with current standard practice of non-surgical management among older people presenting to hospital with pubic rami and concomitant sacral fractures.

## METHODS

A single-site, parallel, two-arm randomised controlled feasibility trial with participants allocated to either surgical or non-surgical intervention on a 1:1 ratio. Participants aged 70 years and over, ambulating with/without walking aids prior to their injury, admitted within 28 days of their injury and had a type 1 lateral compression (LC) pelvic fracture based on the Young-Burgess classification

were invited to participate. The Young-Burgess classification is based on the predominant direction of the vector force at the time of injury. A type 1 LC fracture involves an oblique or transverse pubic rami fracture and ipsilateral sacral compression fracture.[25] Fractures were confirmed either by CT or MRI. In the event of bilateral fractures, participants fulfilling the rest of the eligibility criteria would still be eligible for recruitment. Exclusion criteria were complex pelvic fractures (eg, fractures involving/or close to the hip joint) requiring urgent surgery or progressive weight bearing exercises, pathological fracture in the context of known or suspected malignancy, previous surgery to the pelvis, any condition that precludes surgery or general/spinal anaesthesia, bedbound prior to the injury, receiving palliative care and clinically moribund on admission. During the start of the study, patients with a fracture that had occurred more than 5 days before hospital admission were also excluded. This was later amended to 28 days.

Participants had baseline data collected on recruitment and follow-up assessments at weeks 2, 4 and 12 post-randomisation. All follow-ups were done via a telephone interview except for week 2 where a face-to-face interview was conducted. Data were collected to assess the feasibility of this study and outcome measures for a future definitive trial. For the feasibility outcomes, information was gathered on the number of eligible patients, number of patients and doctors willing to be randomised, adherence to randomisation, rate of participant recruitment and retention, data on the completeness and variability of definitive trial outcome measures, failure of non-surgical care and adverse events in both arms. Outcome data collected for the definitive trial included: the timed up and go test (TUG),[26] Roland Morris Disability Questionnaire (RMDQ),[27] Montreal Cognitive Assessment,[28] Functional Independence Measure,[29] Clinical Frailty Scale,[30] Charlson Comorbidity Index,[31] Barthel Activities of Daily Living Index,[32] Numeric Pain Rating Scale[33] and EuroQoL 5 Dimensions (EQ-5D-3L) score.[34]

Participants were randomly allocated to either surgical intervention or non-surgical care (control group) via a secure web-based system (Sealed Envelope) by a member of the research team after completion of baseline data collection. The surgical team were informed of each participant's allocation. Those randomised to have surgery were assessed by a member of the surgical team for their suitability and choice of surgery based on the participant's general condition, fracture characteristics and surgeon's preference or experience. All surgeries were planned to be carried out within 7 days post-randomisation. Pending surgery, participants received analgesia and had the required preoperative tests. Participants randomised to the non-surgical arm would be started on appropriate analgesia and titrated accordingly. They also had input from the wider multidisciplinary team. If the participant's responsible medical team deemed there was a lack of response to non-surgical treatment, they could refer the participant to be considered for surgery. Participants who

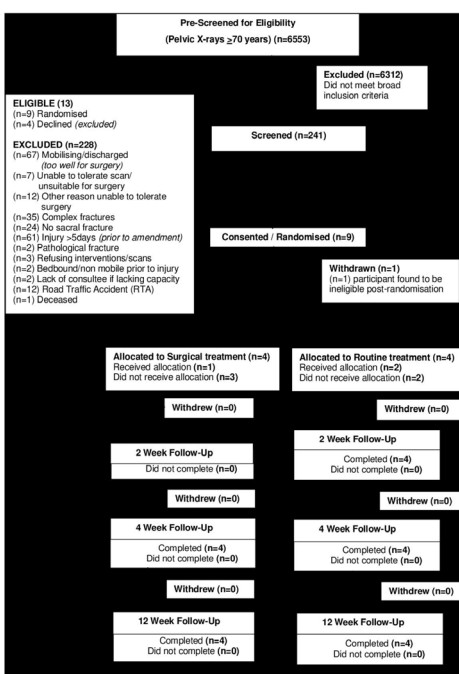

**Figure 1** Consolidated Standards of Reporting Trials diagram for the study.

responded to analgesia while waiting for surgery would also have their indication for surgery reassessed.

Sample size was calculated using data from another UK hospital of its pelvic fracture numbers.[8] A 10-month recruitment period was proposed, with the expectation to screen approximately 100 patients. Taking into account the assumption that 20% of patients screened would be ineligible, and that a 60% recruitment rate would be achieved during the recruitment period, it was then planned that a total of 48 participants would be recruited into the study. Furthermore, with an assumed 10% 3-month attrition rate, it was estimated that 43 participants would complete the study. If follow-up had been completed for these participants, it would have allowed the SD of the TUG to be estimated with an approximate SE of 1.2 assuming the SD is approximately 8 (95% CI: 6.6 to 10.2) and an SE of 0.9 for the RMDQ, assuming the SD is about 6 (95% CI: 4.9 to 7.6).

Participant characteristics and outcome data were reported using appropriate descriptive statistics by treatment arm and overall. The feasibility outcomes were also analysed descriptively. Outcomes were analysed on an intention-to-treat basis. The study was registered on a clinical trials registry). The full protocol has been published.[35] Reporting of this study adhered to Consolidated Standards of Reporting Trials reporting guidelines.

### Patient and public involvement

This study received patient and public involvement (PPI) input through volunteer members of the Royal Osteoporosis Society's local support group. This study's PPI members had personal experience of PFFs and were included in the grant application. Focus group discussions with members of the local support group were also

conducted which informed the design of the study and choice of study outcomes for the trial. All participant-facing documents were reviewed by PPI members. The PPI members were members of the Trial Management Group.

## RESULTS

A total of 241 potential participants were screened over the recruitment period from 15 November 2018 to 31 July 2019. Among those screened, 13 (5.4%) were deemed eligible to take part in the study. The most frequent reasons for exclusion were because participants where either able to mobilise or had discharge plans made already (n=67), participants with complex fractures (n=35), participants with no sacral fracture (n=24), as well as participants whose injury occurred more than 5 days before their hospital admission (n=61, prior to amendment to eligibility criteria) (figure 1).

Of the 13 eligible participants, 9 (69.2%) consented to take part in the study (figure 1). These participants sustained a combination of pelvic and sacral fractures after a fall from a standing height or less. A total of six participants randomised into the study had acute medical issues in addition to their PFF.

Five participants were randomised to the surgical treatment group and four to the non-surgical treatment group. One participant allocated to the surgical treatment group was subsequently withdrawn before receiving their allocated treatment as an exclusion criterion was identified post-randomisation. Four participants were allocated to each intervention group (table 1). The clinical team and spinal surgical team were willing to randomise and adhere to the participant's treatment allocation. After subsequent assessments, only one participant (20%) in the surgical treatment group and two participants (50%) in the non-surgical treatment group received the allocated intervention.

Demographic, baseline characteristics and procedural information of the participants recruited are detailed in table 1.

A total of three participants randomised into the study received surgical treatment regardless of their treatment allocation (one participant in the surgical treatment group and two participants in the non-surgical treatment group). The overall median time to operation was 6 days. All participants had cement augmentation. Data on any screws used were not available. Intraoperatively, one participant reported cement leakage and another one developed a respiratory problem.

The overall median (IQR) length of hospital stay corresponding to the eight participants taking part in the study was 10 (4.5–19.5) days for those in the surgical treatment group and 7 (5.0–23.0) days for those in the non-surgical treatment group. Of the four participants in the surgical treatment group, two (50%) were discharged home without support and the remaining two (50%) to a rehabilitation facility. With regard to those in the non-surgical

**Table 1** Demographics, baseline characteristics and procedural information of the participants recruited

| Characteristics | Surgical treatment group (n=5) | Non-surgical treatment group (n=4) |
|---|---|---|
| Age, median (IQR) years | 85 (83–88) | 85.5 (84–89.5) |
| Female, n (%) | 5 (100) | 4 (100) |
| Charlson Comorbidity Index, median (IQR) | 0 (0–1) | 0.5 (0–1) |
| Montreal Cognitive Assessment, median (IQR) | 23 (16–23) | 24 (22–29)* |
| Clinical Frailty Scale, median (IQR) | 6 (4–6)† | 3 (2.5–5) |
| Prescribed strong opioids, n (%) | 5 (100) | 4 (100) |
| Concomitant acute medical issues, n (%) | 4 (80) | 2 (50) |
| Presence of delirium, n (%) | 0 (0) | 1 (25) |

*Data from three participants.
†Data from four participants.

treatment group, one participant (25%) was discharged home with care assistance and the remaining three (75%) to a rehabilitation facility.

The overall proportion of completeness of outcome data collection at weeks 2, 4 and 12 was at least 75%. One participant was unable to take part in all the assessments corresponding to the Numeric Pain Rating Scale and the EQ-5D-3L questionnaires due to cognitive impairment. Clinical outcomes are reported in table 2.

Adverse events collected up to the 12-week follow-up time point were reported in seven out of eight participants (87.5%). None were related to the intervention provided for their fractures.

## DISCUSSION

This feasibility study aimed to determine if a definitive clinical trial examining the role of spinal sacral fixation for sacral fractures and concomitant pubic rami was deliverable, as such a trial had never been conducted before. It was designed to be pragmatic in nature. Its eligibility criteria were inclusive to reflect what would commonly be encountered in clinical practice where the ideal management of these patients remains uncertain. However, the study highlighted the challenges of delivering such a trial on a larger scale. It was unable to recruit adequate participants to meet the planned sample size. Despite active screening, the number of eligible participants who fulfilled the eligibility criteria was just over 5% (13 out of 241 screened).

Among those screened, almost 30% (67 out of 241 screened) were deemed clinically 'too well for surgery' by their medical team. They were able to mobilise with the analgesia prescribed and inpatient rehabilitation delivered by the multidisciplinary team. This echoed what had been reported in existing literature where most patients admitted with such fractures would be non-operatively managed.[19 36] Of those eligible in this study, approximately 31% (4 out of 13 eligible) declined to participate in the study, which was within what the study had anticipated. Although the treating clinicians and the surgical

team were willing to randomise and adhere to the participants' allocated treatment, not all participants ultimately received the treatment they were allocated to. There were participants allocated to the surgical group where either their pain symptoms improved while waiting for surgery which negated the need for surgery; or on further assessment, the risk of surgery outweighed its potential benefit. The reverse was true for those allocated to the non-surgical group where despite optimal medical care, pain and disability persisted, and they were offered surgery.

Expanding the inclusion criteria to recruit those with only a sacral fracture could have potentially increased the number of participants recruited into the study. There may also have been patients with an acute pubic rami fracture but not had any further imaging done of the pelvis to detect further injuries. Only 46% of those admitted to hospital with a public rami fracture seen on plain radiograph underwent further imaging to visualise the entire pelvis.[37] At least half of pubic rami fractures have a concomitant posterior pelvic fracture but unless suspected by the clinician, this would either be missed or diagnosed late,[7] thus, missing potential participants. Hence, an important requirement for such trials in the future is to embed detailed pelvic imaging in patients with a confirmed pubic rami fracture. However, an argument could be made that if patients were already improving and becoming less symptomatic following their fracture, further imaging would be unlikely to alter the treatment plan in clinical practice. Of the participants recruited, most were able to provide outcome measures for the required domains. Some assessment was limited by the presence of cognitive impairment. All participants were able to adhere to the follow-up schedule.

This study was not designed to look at the effectiveness of surgical intervention compared with medical care. The data available were also unable to determine any trends or significant differences in outcomes between groups. However, this does not necessarily mean that there is no role for surgical intervention for older patients with these fractures. The non-randomised studies to date

**Table 2** Outcomes at follow-up visits compared with baseline measurement

| Characteristics: median (IQR) | Surgical treatment group (n=4) | | | | Non-surgical treatment group (n=4) | | | |
|---|---|---|---|---|---|---|---|---|
| | Baseline | Week 2 | Week 4 | Week 12 | Baseline | Week 2 | Week 4 | Week 12 |
| Timed up and go test, measured in seconds | – | 47.2 (29.9–88.6) | – | 22.6 (16.7–25.1) | – | 53.7 (28.0–210.0)* | – | 19.9 (19.0–47.8)* |
| Roland Morris Disability Questionnaire | – | 13 (11.0–15.5) | 14.5 (9.5–15.5) | 8.5 (4.5–10.5) | – | 17.0 (14.0–22.0)* | 12.0 (10.0–20.0)* | 10.5 (6.0–14.0) |
| Functional Independence Measure | 77.5 (67.5–88.0) | 114 (91–119)* | – | 120.5 (114.5–125.0) | 77.0 (51.5–92.5) | 100 (57–117) | – | 115.0 (86.0–120.0) |
| Barthel Activities of Daily Living | 11 (9–13) | 15.0 (11.5–18.0) | 19 (16–19)* | 19.0 (18.5–19.5) | 9 (5.5–14.0) | 14.0 (7.0–17.0) | 18.0 (14.0–19.0)* | 18.5 (11.5–20.0) |
| Numeric Pain Rating Scale | 10 (9–10) | 5 (4.0–7.0)* | 5.0 (3.5–6.5) | 4.5 (2.5–6.0) | 10 (8–10)* | 7 (6.0–8.0)* | 7.0 (3.0–9.0)* | 4.5 (4.0–5.0)† |

*Data from three participants.
†Data from two participants.

have suggested a role for surgery in improving symptoms.[12 22–24 36] Additionally, the number of older people sustaining PFFs will increase and alongside it the healthcare utilisation to support them back to recovery. Surgical treatment may have a role in optimising recovery, similar to the role hip fracture fixation has in getting patients out of bed as early as the next day.[37] Hence, there is a need for randomised clinical trials to inform clinicians the likely role of surgery in PFFs. To date, it remains uncertain what patient, clinical or fracture characteristics that would benefit from surgery. A pelvic fracture specialist group have also put forward a different classification for pelvic fractures specifically for older people with low trauma pelvic fractures to support better stratification of patients for surgical or non-surgical management.[38]

Hence, clinical trials are clearly required to understand the role, its effectiveness and timing of surgery in this group of patients. This was the first study that has looked at how best to design a trial to evaluate this. Issues around participant identification, eligibility, recruitment and understanding treatment decisions of hospital care still need to be addressed before a definitive trial. Such an approach where a feasibility study is conducted before a definitive trial is becoming more common.[39] Feasibility studies with clear objectives of what aspect is being investigated, such as recruitment capability, data and outcome collection procedures, acceptability and suitability of the intervention or study procedure, evaluation of the resources to deliver the study and participants' response to the intervention, improve the design of a future trial.[40] This study was an important first study in defining the parameters for a definitive, complex trial. Moving forward, addressing the recruitment challenges identified here is needed. A single hospital site will not be able to achieve the required numbers. A multisite-centre study is needed. Creating a network of hospitals that provide pelvic fracture surgery in the UK may support delivering the numbers required for a definitive trial.

**Author affiliations**
¹Department of Healthcare for Older People, Nottingham University Hospitals NHS Trust, Nottingham, UK
²Department of Medicine, Faculty of Medicine, Universiti Malaya, Kuala Lumpur, Malaysia
³Leicester Clinical Trials Unit, University of Leicester, Leicester, UK
⁴Division of Rehabilitation and Ageing, School of Health Sciences, University of Nottingham, Nottingham, UK
⁵School of Health Sciences, Faculty of Medicine & Health Sciences, University of Nottingham, Nottingham, UK
⁶School of Medicine, University of Nottingham, Nottingham, UK
⁷Centre for Spinal Studies and Surgery, Nottingham University Hospitals NHS Trust, Nottingham, UK

**Acknowledgements** The study team would like to acknowledge the input from the Healthcare of Older People research team who were involved in screening, recruitment and follow-up of the study participants. The team would also want to thank the patient and public involvement representatives, members of the Trial Steering Committee, Leicester Clinical Trials Unit, staff from the Department for Healthcare of Older People and Department for Spinal Surgery at Nottingham University Hospitals NHS Trust.

**Contributors** Conceptualisation—TO, ASDP, CB, AD, PH, PL, MJ, KS, NQ and OS. Methodology—TO, ASDP, CB, AD, PH, PL, MJ, KS, NQ and OS. Formal analysis—TO, ASDP, CB, AD and OS. Investigation—TO, KS, NQ and OS. Data curation—ASDP and CB. Writing (original draft preparation)—TO, AD and OS. Writing (review and editing)—TO, ASDP, CB, AD, PH, PL, MJ, KS, NQ and OS. Project administration—OS. Funding acquisition—TO, ASDP, CB, AD, PH, PL, MJ, KS, NQ and OS. Data guarantor - OS

**Funding** This study was funded by the National Institute for Health Research (NIHR) under its Research for Patient Benefit (RfPB) Programme (grant reference number PB-PG-0816-20002).

**Competing interests** None declared.

**Patient and public involvement** Patients and/or the public were involved in the design, or conduct, or reporting, or dissemination plans of this research. Refer to the Methods section for further details.

**Patient consent for publication** Not required.

**Ethics approval** This study involves human participants and was approved by North East, Newcastle and North Tyneside 2 research ethics committee (reference number 18/NE/0212). Participants gave informed consent to participate in the study before taking part.

**Provenance and peer review** Not commissioned; externally peer reviewed.

**Data availability statement** Data are available upon reasonable request. The datasets generated during and/or analysed during the current study are/will be available upon request from the Chief Investigator Professor Opinder Sahota ( Opinder.Sahota@nuh.nhs.uk).

**ORCID iDs**
Terence Ong http://orcid.org/0000-0001-7473-446X
Avril Drummond http://orcid.org/0000-0003-1220-8354
Opinder Sahota http://orcid.org/0000-0003-0055-7637

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
