## [Reviewer comments · BMJ Open]

ARTICLE DETAILS

TITLE (PROVISIONAL)	The ASSERT (Acute Sacral insufficiency fracture augmentation) Randomised Controlled, Feasibility Trial in Older People
AUTHORS	Ong, Terence; Suazo Di Paola, Ana; Brookes, Cassandra; Drummond, Avril; Hendrick, Paul; Leighton, Paul; Jones, Matthew; Salem, Khalid; Quraishi, Nasir; Sahota, Opinder

VERSION 1 – REVIEW

REVIEWER	Kim, Hyeun Sung Nanoori Hospitals
REVIEW RETURNED	24-May-2021

GENERAL COMMENTS	This study is a comparative study of surgical and non-surgical acute sacral insufficiency fracture. However, because the number of comparison groups is too small and the results and conclusions are not clear, it seems difficult to clearly determine the purpose of this study.
--

REVIEWER	Rommens, PM University Medical Center Mainz, Mainz, Germany, Department of Orthopedics and Traumatology
REVIEW RETURNED	12-Nov-2021

GENERAL COMMENTS	Interesting paper on the feasibility of a randomized trial comparing conservative versus operative treatment of fragility fractures of the pelvis. Huge work has been done on this topic from the “pelvic working group” of the Department of Orthopaedics and Traumatology of the University Medical Center in Mainz, Germany. See references below. None of this work has been cited (with the exception of reference 19, which is one of their earliest publications). One would expect that at least some of their publications are cited in this new publication (in the introduction and/or discussion). Especially the manuscript on classification and recommendation of treatment (from 2013) must be regarded as a key-stone publication. Please insert some of these publications and comment. 1. Rommens PM, Hofmann A. Comprehensive classification of fragility fractures of the pelvic ring: Recommendations for surgical treatment. Injury. 2013 Dec;44(12):1733-44. doi: 10.1016/j.injury.2013.06.023. Epub 2013 Jul 18. PMID: 23871193. 2. Rommens PM, Wagner D, Hofmann A. Fragility Fractures of the Pelvis. JBJS Rev. 2017 Mar 21;5(3):e3. doi: 10.2106/JBJS.RVW.16.00057. PMID: 28359073. 3. Wagner D, Kisilak M, Porcheron G, Krämer S, Mehling I, Hofmann A, Rommens PM. Trans-sacral bar osteosynthesis provides low mortality and high mobility in patients with fragility fractures of the
---

pelvis. *Sci Rep*. 2021 Jul 9;11(1):14201. doi: 10.1038/s41598-021-93559-0. PMID: 34244526; PMCID: PMC8270908.

4. Rommens PM, Hopf JC, Herteleer M, Devlieger B, Hofmann A, Wagner D. Isolated Pubic Ramus Fractures Are Serious Adverse Events for Elderly Persons: An Observational Study on 138 Patients with Fragility Fractures of the Pelvis Type I (FFP Type I). *J Clin Med*. 2020 Aug 3;9(8):2498. doi: 10.3390/jcm9082498. PMID: 32756494; PMCID: PMC7463797.

5. Wagner D, Hofmann A, Kamer L, Sawaguchi T, Richards RG, Noser H, Gruszka D, Rommens PM. Fragility fractures of the sacrum occur in elderly patients with severe loss of sacral bone mass. *Arch Orthop Trauma Surg*. 2018 Jul;138(7):971-977. doi: 10.1007/s00402-018-2938-5. Epub 2018 Apr 26. PMID: 29700604.

6. Wagner D, Kamer L, Sawaguchi T, Richards RG, Noser H, Rommens PM. Sacral Bone Mass Distribution Assessed by Averaged Three-Dimensional CT Models: Implications for Pathogenesis and Treatment of Fragility Fractures of the Sacrum. *J Bone Joint Surg Am*. 2016 Apr 6;98(7):584-90. doi: 10.2106/JBJS.15.00726. PMID: 27053587.

7. Rommens PM, Wagner D, Hofmann A. Do We Need a Separate Classification for Fragility Fractures of the Pelvis? *J Orthop Trauma*. 2019 Feb;33 Suppl 2:S55-S60. doi: 10.1097/BOT.0000000000001402. PMID: 30688861.

8. Rommens PM, Hofmann A, Kraemer S, Kisilak M, Boudissa M, Wagner D. Operative treatment of fragility fractures of the pelvis: a critical analysis of 140 patients. *Eur J Trauma Emerg Surg*. 2021 Oct 11. doi: 10.1007/s00068-021-01799-6. Epub ahead of print. PMID: 34635938.

Page 6/30, line 56: pubic ramus and concomitant sacral fractures may exhibit different degrees of pelvic instability. See the new classification, published in 2013. It would be wise to restrict the study to a specific subgroup of fragility fractures of the pelvis (FFP). FFP type II (which is equal to a low-energy lateral compression fracture) is the most frequent subtype and its treatment remains controversial. FFP type I are treated conservatively and surgery is recommended for FFP types III and IV. It therefore makes sense to choose FFP type II for such a study.

Page 7/30, line 10: LC type I - were bilateral pubic rami or bilateral sacral fractures excluded?

Page 7/30, line 30: the ultimate goals of treatment of FFP pain relief and recovery of mobility. Why didn't you use the Parker Mobility Score, which focuses on mobility only and is very easy to capture.

Page 7/30, line 40: it is surprising that randomisation took place before suitability of the patient for surgery has been controlled. Wouldn't it be more logical to perform randomisation after informed consent of the study?

Page 8/30, line 24: 67 patients were able to mobilize or had discharge plans already. Does this mean that part of them had been randomized for surgery but refused thereafter because of a favourable progress of pain relief and mobility? It seems that patients (or care providers) were biased against any surgical treatment. Please comment.

Page 9/30, line 32: "too well for surgery". In the retrospective study of Rommens et al,

9. Rommens PM, Boudissa M, Krämer S, Kisilak M, Hofmann A, Wagner D. Operative treatment of fragility fractures of the pelvis is connected with lower mortality. A single institution experience. *PLoS One*. 2021 Jul 9;16(7):e0253408. doi: 10.1371/journal.pone.0253408. Erratum in: *PLoS One*. 2021 Sep

	27;16(9):e0258076. PMID: 34242230; PMCID: PMC8270175. only one fourth of the patients with FFP type II (low-energy LC type I) received surgical treatment. This indicates that not every patient needs surgery. For a prospective randomized study, it would be better to select a subgroup of patients, who do not profit from analgesia during the first days of their hospital stay. In this group, it would be very interesting to experience if surgery makes a difference on mid-term follow up. Page 9/30, line 60 and page 10/30 lines 1-5: The non-randomised studies to date have suggested a role for surgery in improving symptoms. See most recent publications of Rommens et al (9) and of and Wagner et al (3) and comment. The study clearly shows that such a study should be done in multiple hospitals simultaneously. The inclusion criteria for randomization should be adjusted. Probably, FFP type II patients, who do not immediately respond to analgesia (after 3-4 days) are the best candidates for inclusion in such a study.
--	--

REVIEWER	McArthur, John University Hospitals Coventry and Warwickshire NHS Trust
REVIEW RETURNED	22-Nov-2021

GENERAL COMMENTS	Thankyou for submitting this well designed study assessing the feasibility of a RCT investigating surgery for LC1 pelvic fragility fractures. The introduction is clear and highlights the extent of the problem and the limited evidence that exists supporting either surgical or non-surgical management of these common injuries. The study method is appropriate and well described. The eligibility criteria were broad at the start of the trial but were made even broader with an amendment to allow patients presenting up to 28 days rather than 5 days after injury. 61 patients were excluded for this reason prior to this amendment, however, it seems likely that a large number of this 61 would have subsequently been excluded for other reasons such as 'too fit for surgery'. The data are easy to interpret and are clearly presented. The report is very candid about the difficulties that were experienced in trying to recruit this patient group to a RCT. Recruitment numbers were well below those expected (even with broad eligibility criteria) and these problems are likely to be experienced in any further similar trials. It is interesting that the majority of the randomised patients did not receive their allocated treatment. It is clear from the paper that this happened due to clinical judgement rather than lack of surgeon equipoise i.e. assessed as either too mobile for surgery or in too much pain for non-op in the post-randomisation period. I therefore wonder if a few days 'trial of non-operative management' to establish that these patients were definitely 'bad enough for surgery' would have been useful prior to randomisation as has been performed in similar trials. The authors have suggested a further feasibility study to address the problems identified in this study. I would be grateful if they could
---

	elaborate on what this study should address in terms of its aims and methods. Particularly as they suggest involving further centres and my understanding is that this procedure is not widely used for this injury in the UK. Alternatively, I think it could be appropriate to conclude that this feasibility study has proven that undertaking a full scale RCT to answer this question is not possible. Overall I think this study adds a lot of useful information. It gives us evidence of the natural history of these injuries and that the majority do well with non-op management. It also informs us that if we were to try and undertake a full scale RCT to compare the outcomes of the two treatments, we would encounter significant problems with recruitment and crossover, particularly if a similar study design was used.
--	---

VERSION 1 – AUTHOR RESPONSE

Reviewer 1		
The number in the comparison groups is too small and the results and conclusions are not clear, it seems difficult to clearly determine the purpose of this study.	We agree with the reviewer that it was not possible to make any conclusion on the effectiveness of surgical intervention for sacral insufficiency fractures, based on our study. However, our aim was not to determine efficacy but to determine whether a definitive trial was feasible or not. We believe our results do support the design of a future trial to investigate this.	No change
Reviewer 2		
Inclusion of published work from the “pelvic working group” of the Department of Orthopaedics and Traumatology of the University Medical Center in Mainz, Germany	We thank the reviewer for the suggestion. We have now included two recent studies into the manuscript – Rommens PM, et al. J Orthop Trauma. 2019, and Rommens PM, et al. Eur J Trauma Emerg Surg. 2021.	Discussion section, Paragraph 2 and 4.
Pubic ramus and concomitant sacral fractures may exhibit different degrees of pelvic instability. See the new classification, published in 2013. It would be wise to restrict the study to a specific subgroup of fragility fractures of the pelvis (FFP). FFP type II (which is equal to a low-energy lateral compression fracture) is the most frequent subtype and its	This is an insightful comment and we thank the reviewer for this suggestion. We have now included a sentence referencing Rommens PM, Eur J Trauma Emerg Surg, 2021 suggesting that this classification may have a more relevant role in the management of pelvic fragility fractures.	Discussion section, Paragraph 4.

treatment remains controversial. FFP type I are treated conservatively and surgery is recommended for FFP types III and IV. It therefore makes sense to choose FFP type II for such a study.		
LC type I - were bilateral pubic rami or bilateral sacral fractures excluded?	In the event of bilateral fractures, participants would still be recruited if they fulfilled the rest of the eligibility criteria.	Methods section, Paragraph 1. This sentence has been included.
The ultimate goals of treatment of FFP pain relief and recovery of mobility. Why didn't you use the Parker Mobility Score, which focuses on mobility only and is very easy to capture.	We are grateful for this suggestion which we will consider moving forwards. A number of scales were considered and part of the aim of this study was to assess how feasible it was to capture the data from the scales/scores selected.	No change
It is surprising that randomisation took place before suitability of the patient for surgery has been controlled. Wouldn't it be more logical to perform randomisation after informed consent of the study?	This feasibility study was pragmatic. In clinical practice, the dilemma is whether a patient should be operatively or non-operatively managed. Hence, it made sense to randomise at this decision making junction. Randomisation happened after patient consented to take part in the study. They were informed of this, and adherence to operative or non-operative arm was part of our outcome measure.	No change
67 patients were able to mobilize or had discharge plans already. Does this mean that part of them had been randomized for surgery but refused thereafter because of a favourable progress of pain relief and mobility? It seems that patients (or care providers) were biased against any surgical treatment. Please comment.	We apologise for the confusion. These 67 patients were within the 241 patients with PFF screened. They were not recruited into the study at this point. In this group, the treating clinicians reported that they were able to mobilise and were not being considered for an operation. We are unable to comment if the clinicians were biased against surgical treatment but all the clinicians involved were briefed on this study and its aims and were supportive of the study. We also acknowledge that most pelvic fractures would be managed non-operatively.	No change
For a prospective randomized study, it would be better to select a subgroup of patients, who do not profit from analgesia during the first days of their hospital	This is a good suggestion which we will definitely take on board in future study designs. For this study, we wanted to design a feasibility study which would	No change

stay. In this group, it would be very interesting to experience if surgery makes a difference on mid-term follow up.	inform a future trial on early surgical intervention for PFFs. Hence, the recruitment early on in the admission process. As we have seen from hip fracture trials, early intervention consistently makes a difference in patients' outcomes. In our local experience, there is usually a gap of a few days between decision to operate and the day of surgery due to resource and capacity constraints. In this study, the median time to operation was 6 days. During that time usual non-operative care with pain relief was administered.	
The non-randomised studies to date have suggested a role for surgery in improving symptoms. See most recent publications of Rommens et al and of and Wagner et al and comment.	We have decided to kindly decline on commenting on these 2 papers. The focus of this study and manuscript was to design a robust RCT to evaluate the effectiveness of a surgical approach to PFF, and not the merits of surgery in PFF per se. There are reviews already on this topic. This was beyond the remit of our study.	No change
The study clearly shows that such a study should be done in multiple hospitals simultaneously. The inclusion criteria for randomization should be adjusted. Probably, FFP type II patients, who do not immediately respond to analgesia (after 3-4 days) are the best candidates for inclusion in such a study.	As mentioned in the earlier points, we welcome these suggestions moving forward. However this was an early feasibility study. Of note, the median time to operation (Results section) was 6 days. In that time, the patient and clinical team could decide against surgery if they had appropriately responded to a non-operative approach.	No change
Reviewer 3		
I therefore wonder if a few days 'trial of non-operative management' to establish that these patients were definitely 'bad enough for surgery' would have been useful prior to randomisation as has been performed in similar trials.	Thank you for this which was also raised by another reviewer. We also wanted to focus on the role of early surgical intervention. Findings from hip fracture treatment has consistently highlighted the importance of early surgery to facilitate early mobilisation. In our experience, there is also a gap between decision to operate and the date of the operation due to capacity issues. In this study, the median time to operation was 6 days during this time, patients would	No change

	continue with their analgesia and, if the patient or clinician felt that those allocated to surgery no longer required it, they would opt out.	
The authors have suggested a further feasibility study to address the problems identified in this study. I would be grateful if they could elaborate on what this study should address in terms of its aims and methods. Particularly as they suggest involving further centres and my understanding is that this procedure is not widely used for this injury in the UK. Alternatively, I think it could be appropriate to conclude that this feasibility study has proven that undertaking a full scale RCT to answer this question is not possible.	We thank the reviewer for this comment. We recognised the recruitment challenge in undertaking this study. These fractures are not common compared to other fragility fractures such as hip or vertebral fractures. The non-randomised studies have reported treatment benefit with surgery. A RCT would be able to shed light on the effectiveness of the intervention. Creating a network of such centres in the UK that offer this intervention may support delivering the numbers required for such a study.	Discussion section, paragraph 5